# Antioxidant and Cytotoxic Effects on Tumor Cells of Exopolysaccharides from *Tetraselmis suecica* (Kylin) Butcher Grown Under Autotrophic and Heterotrophic Conditions

**DOI:** 10.3390/md18110534

**Published:** 2020-10-26

**Authors:** Geovanna Parra-Riofrío, Jorge García-Márquez, Virginia Casas-Arrojo, Eduardo Uribe-Tapia, Roberto Teófilo Abdala-Díaz

**Affiliations:** 1Doctorado en Acuicultura, Programa Cooperativo Universidad de Chile, Universidad Católica del Norte, Pontificia Universidad Católica de Valparaíso, Valparaíso 2340000, Chile; 2Departamento de Acuicultura, Facultad de Ciencias del Mar, Universidad Católica del Norte, Larrondo 1281, Coquimbo, Chile; euribe@ucn.cl; 3Department of Microbiology, Faculty of Sciences, University of Malaga, 29071 Malaga, Spain; j.garcia@uma.es; 4Instituto de Biotecnología y Desarrollo Azul (IBYDA), Departamento de Ecología y Geología, Facultad de Ciencias, Universidad de Málaga, 29071 Málaga, Spain; virginiac@uma.es

**Keywords:** *Tetraselmis suecica*, autotrophic culture, heterotrophic culture, exopolysaccharides, antioxidant capacity, cytotoxic effects on tumor cells

## Abstract

Marine microalgae produce extracellular metabolites such as exopolysaccharides (EPS) with potentially beneficial biological applications to human health, especially antioxidant and antitumor properties, which can be increased with changes in crop trophic conditions. This study aimed to develop the autotrophic and heterotrophic culture of *Tetraselmis suecica* (Kylin) Butcher in order to increase EPS production and to characterize its antioxidant activity and cytotoxic effects on tumor cells. The adaptation of autotrophic to heterotrophic culture was carried out by progressively reducing the photoperiod and adding glucose. EPS extraction and purification were performed. EPS were characterized by Fourier-transform infrared spectroscopy and gas chromatography-mass spectrometry. The antioxidant capacity of EPS was analyzed by the 2,2’-azino-bis (3-ethylbenzothiazoline-6-sulphonic acid) (ABTS) method, and the antitumor capacity was measured by the 3-(4,5-dimethylthiazol-2-yl)-2,5-diphenyltetrazolium bromide (MTT) assay, showing high activity on human leukemia, breast and lung cancer cell lines. Although total EPS showed no cytotoxicity, acidic EPS showed cytotoxicity over the gingival fibroblasts cell line. Heterotrophic culture has advantages over autotrophic, such as increasing EPS yield, higher antioxidant capacity of the EPS and, to the best of our knowledge, this is the first probe that *T. suecica* EPS have cytotoxic effects on tumor cells; therefore, they could offer greater advantages as possible natural nutraceuticals for the pharmaceutical industry.

## 1. Introduction

Microalgae can develop in autotrophic, mixotrophic and heterotrophic conditions due to their physiological characteristics and plasticity of adaptation on our planet [1], differentiating them by the type of energy and the source of carbon to be used [2]. The cultivation of microalgae is mostly performed in autotrophy using light as an energy source, which is transformed by photosynthesis into chemical energy for the storage of polysaccharides and lipids [3,4].

In a heterotrophic culture, microalgae have the ability to grow and metabolize organic carbon sources with limited irradiance [5], changing their metabolism to generate energy by breathing or using an organic substrate under heterotrophic conditions [6]. It has several advantages over an autotrophic crop: (i) it does not require lighting, (ii) high biomass yield, (iii) high growth rates and (iv) increased synthesis of metabolites of scientific and biotechnological interest [5,6,7]. These metabolites depend on changes in culture media; therefore, optimization of organic carbon, macronutrient and micronutrient concentrations is sought in order to obtain the best yields in terms of productivity and biomass [1,8].

Crop types and systems provide advances both for the production of biomass with high nutritional value and economic viability and for the search, extraction and characterization of new products [9,10]. Heterotrophic culture has a potential market for the production of high-value metabolites with respect to autotrophic crops: (i) lipids with four times higher content of polyunsaturated fatty acids (PUFAs) than those cultured in autotrophic conditions [11,12,13], (ii) accumulation of up to 45% by dry weight of carbohydrates [11,14,15], (iii) a higher percentage of proteins for the biomass [16] and (iv) pigments (lutein and phycocyanin) with balanced C/N ratios and astaxanthin with limiting nitrogen [17,18,19].

*Tetraselmis suecica* is a marine green microalga widely used in aquaculture as live food for rotifers and copepods or Artemia in hatcheries [20]. This microalga can be cultivated in autotrophic and heterotrophic conditions [21]. It has antibacterial activity [22,23], probiotic properties [24] and has been proposed as a source of vitamin E for humans [25]. *T. suecica* produces exopolysaccharides [26]; however, their structural characteristics and biotechnological applications in human health remain unknown.

Polysaccharides are high-molecular weight molecules that contain repetitive structural units—monosaccharides- joined by glucosidic bonds, forming linear or branched structures. This structural variability has biotechnological interest [27]. Its applications are promising due to its immunomodulatory, antimicrobial, antiviral, antioxidant and antitumoral capacities [28]. Algal polysaccharides are free radical scavengers and, therefore, have antioxidant effects and prevent oxidative damage in living organisms [29]. The antioxidant activity of polysaccharides has been related to the presence of sulfates and uronic acids in them [28,30]. In this sense, in vitro antiproliferative activity in human cancer cells lines [31] and in vivo inhibition of Graffi myeloid tumor growth in hamsters [32] have been demonstrated with marine algal polysaccharides. Therefore, the prospecting of natural-origin compounds such as polysaccharides is a source for the prevention of diseases that counteract the toxic effects of synthetic compounds.

The aim of this work was to develop the crop of *T. suecica* under autotrophic and heterotrophic conditions, comparing the differences between the biochemical composition of the algal biomass and their yield exopolysaccharides (EPS). Furthermore, the functional groups, monosaccharides characterization and the antioxidant activity and cytotoxic effects on tumor cells and healthy cells of the exopolysaccharides were assessed.

## 2. Results and Discussion

### 2.1. Adaptation of Autotrophic to Heterotrophic Culture of T. suecica

The heterotrophic culture of *T. suecica* showed that cell density, cell concentration and biovolume were statistically higher in the heterotrophic culture (*p* < 0.05), while the cell volume was 17 times lower compared with the autotrophic culture (*p* < 0.05) (Table 1). The specific growth rate between autotrophic and heterotrophic cultures no showed statistic differences significantly (Table 1).

Azma et al. [12] obtained differences in the final cell concentration of *T. suecica* grown in autotrophy and heterotrophy. On the contrary, Day and Tsavalos [33] found no differences in the final cell concentration of *Tetraselmis* sp. between the two culture conditions. These variations could be due to growth in the absence of light, and the presence of organic substrates can change the metabolism and morphology of cells. In our investigation, glucose was used as the source of organic carbon, which generated high cellular concentrations due to the energy provided (2.8 kJ mol^−1^), compared to the 0.8 kJ mol^−1^ for acetate used in Azma et al.’s [12] investigation.

The adaptation of autotrophic to heterotrophic culture was performed by the progressive reduction of the illumination times in the photoperiod, preserving the irradiance of the *T. suecica* cultures. However, Azma et al. [21] made a progressive decrease in lighting for *T. suecica* cultures with longer periods, adding a total of 1650 h compared to the present study, which was 1080 h for adaptation to heterotrophy, meaning 35% less hours of adaptation, which would be due to the different media used in cultivation. The Walne medium [34] used in Azma et al.’s [21] investigation contained concentrations of nitrate, phosphate, ethylenediaminetetraacetic acid (EDTA), zinc, molybdenum and manganese higher than F/2 used in the present study. Therefore, *T. suecica* has the ability to regulate its metabolism to achieve balanced growth in heterotrophic culture; this capability can be used to increase the production of metabolites of biotechnological interest.

### 2.2. Elemental Analysis of Autotrophic and Heterotrophic Biomass Cultures of T. suecica

Heterotrophic cultures of *T. suecica* showed higher carbon and nitrogen contents than those observed in autotrophy (*p* < 0.05; Table 2). The C/N ratio did not show statistical differences (*p* > 0.05) between both culture conditions (Table 2). The C/N ratio is a nutritional indicator of the microalgae. When N is low, the C/N ratio favors the biosynthesis and accumulation of carbohydrates [35]. However, Cheng et al. [36] mentioned that a C/N ratio higher than 10 allowed lipid accumulation in heterotrophic cultures. In our study, the high C/N ratio is favored for the accumulation of carbon, which will be used to increase the accumulation of lipids or carbohydrates.

### 2.3. Biochemical Composition of Autotrophic and Heterotrophic Biomass Cultures of T. suecica

Heterotrophic cultures eliminate the light limitations that autotrophic cultures require, generating metabolic changes in microalgae, thus presenting variations in the biochemical composition of the biomass [6,7]. In our study, except for the percentage of ash (*p* > 0.05; Table 3), statistical differences in the biochemical composition of the autotrophic and heterotrophic cultures of *T. suecica* were found (*p* < 0.05; Table 3).

The percentage of proteins in *T. suecica* statistically increased from 16.76% in autotrophy to 20.78% in heterotrophy (*p* < 0.05; Table 3). To the best of our knowledge, no studies regarding protein increase with respect to heterotrophic cultures has been reported; however, Cid et al. [37] showed that the addition of organic compounds to the culture medium increased the protein fraction in *T. suecica* mixotrophic cultures. El-Sheekh et al. [16] reported a significant increase in the percentage of proteins in mixotrophic cultures of *Chlorella vulgaris* and *Scenedesmus obliquus* with the addition of hydrolyzed wheat bran with respect to autotrophy. Canelli et al. [38] mentioned that heterotrophic cells convert the storage of cellular nitrogen into proteins, and when this nitrogen reserve is depleted, the consumption of intracellular carbon begins to increase the protein fraction. However, it is important to consider that a low C/N ratio induces protein accumulation [39].

In the case of lipids and carbohydrates, an increase in heterotrophic cultures of *T. suecica* is observed with respect to autotrophic cultures (*p* < 0.05; Table 3). In this sense, Azma et al. [12] observed that the heterotrophic culture of T. suecica presented a higher percentage of lipids with respect to the autotrophic culture. Furthermore, similar results were observed in heterotrophic cultures of *Chlorella protothecoides* and *C. vulgaris*, with lipid accumulations between 50–60% in the biomass [11,13]. Similar results of carbohydrate accumulation (>45%) in dry weight were observed for these microalgae species [15]. In our study, *T. suecica* increased in a greater percentage the carbohydrate content in relation to the lipids (Table 3). This could be because there was probably a depletion of N, observed by the high C/N index (Table 2). High growth rates in heterotrophic cultures led to nutrient depletion, decreasing cell division and allow them to accumulate carbon for the synthesis of lipids or carbohydrates [40]. Another factor could be the nitrogen deficiency, which induced the increase of lipids in the biomass [41]. Furthermore, Garcia-Ferris et al. [41] observed that, in periods of nitrogen starvation and heterotrophy, there was a decrease in the size of the chloroplast of *Euglena gracilis*. Additionally, in our study was observed a reduction in chloroplast size in the heterotrophic culture (data not shown). Similar results were observed by Gladue and Maxey [14] for *Tetraselmis* sp. in a heterotrophic culture.

### 2.4. Phenolic Compounds and Antioxidant Activity of Autotrophic and Heterotrophic Biomass Cultures of T. suecica

The phenol content of heterotrophic cultures of *T. suecica* was higher than autotrophic (*p* < 0.05; Table 4). To the best of our knowledge, our results are the first report of phenol content in heterotrophic cultures for the species under study. In heterotrophy, the mechanisms of accumulation and antioxidant response are related to nutritional stress [42]. Phenolic compounds are a defense mechanism against the excess of oxygen produced in photosynthesis, and the depletion of nutrients causes their accumulation [42]. Quiñones-Galvez et al. [43] showed that the heterotrophic cultivation of calluses of *Theobroma cacao* increased the phenolic content, due to the osmotic stress caused by the addition of glucose, favoring their synthesis and accumulation. This increase in phenolic compounds in heterotrophy was also observed for *C. vulgaris* and *S. obliquus* [44].

The increase in antioxidant activity is due to the fact that the photosystem II of the cells produces reactive oxygen species (ROS), caused by the photosynthetic process [45]. However, in the adaptation of autotrophy to heterotrophy, photosystem II is reduced due to its low photosynthetic activity [46], decreasing the chlorophyll, carotene and phycobiliprotein contents, which are related to the nitrogen availability, and causing alterations in the electron transport system, leading to an increase in antioxidant activity [47,48].

The 2,2’-azino-bis (3-ethylbenzothiazoline-6-sulphonic acid) (ABTS) method measures hydrophilic and lipophilic antioxidants [49]. *T. suecica* increased the antioxidant activity measured by the ABTS method in heterotrophic cultures with respect to autotrophic (*p* < 0.05; Table 4). A positive correlation was found in the heterotrophic culture of *T. suecica* between ABTS and proteins, lipids, carbohydrates, total carbon (TC) and total nitrogen (TN) (Appendix A). This could be because the method also measures fat-soluble antioxidants (carotenoid, chlorophylls, vitamin E or tocopherols, PUFAs and polysaccharides) that are part of the biomass [49]. Although, in this study, no analysis of fatty acid composition was performed, an increase in the content of PUFAs in an heterotrophic cultivation of *T. suecica* has been reported [14,20], which would indicate that the increase in fatty acid composition is related to higher antioxidant activity [50].

The 2,2-diphenyl-1-picrylhydrazyl (DPPH) method measures the reducing capacity of the hydrophilic fraction of the compound [51]. In our study, *T. suecica* increased the antioxidant capacity in heterotrophic cultures with respect to autotrophic measured by the DPPH method (*p* < 0.05; Table 4). A positive correlation was found in the heterotrophic culture of *T. suecica* between the DPPH and phenolic content, lipids and TC (Appendix A). Therefore, it is attributed that the increase in antioxidant activity by the DPPH method is related to the content of phenols, because this assay performs a better measurement of hydrophilic compounds. Significant correlations have been observed in macroalgae [52]; however, in microalgae so far has not been found a correlation between the phenol content and DPPH, so this study shows the first evidence for a heterotrophic culture of *T. suecica*. Other authors differ from this relation, indicating that the variety of specific phenolic compounds in microalgae must be understood to which these correlation differences are attributed [48,53]. However, a synergistic effect among other compounds or substances could be involved in the antioxidant activity of microalgae, so future research would focus on correlating the increase in antioxidant activity in heterotrophic cultures with other variables involved in this type of condition.

### 2.5. Pigment Content of Autotrophic and Heterotrophic Biomass Cultures of T. suecica

The heterotrophic culture of *T. suecica* reduced chlorophyll and carotenoid levels to 1% and 12%, respectively (*p* < 0.05), with respect to that observed in autotrophy (Figure 1). Our results are in-line with those by Day and Tsavalos [33], who reported a reduction of chlorophyll levels to 1% and carotenes to 50% of *T. suecica* in heterotrophic culture, changing the cells from green to bright yellow, an adaptation caused by the absence of light and observed in higher plants [54].

The reduction of photosynthetic and auxiliary pigments is related to the absence of light in heterotrophic cultures and to nitrogen depletion [41,55]. The stress generated by the changes in the trophic conditions and nutrients must be evaluated to identify potential microalgae that may produce some pigment of interest under dark conditions.

### 2.6. Production and Extraction of Exopolysaccharides (EPS) of Autotrophic and Heterotrophic Biomass Cultures of T. suecica

The maximum concentration of total and acid exopolysaccharides (EPS) extracted from the heterotrophic culture of *T. suecica* was 4.2 and 8 times higher, respectively, with respect to that obtained in the autotrophic culture (*p* < 0.05; Figure 2).

The polysaccharide production for *T. suecica* has mainly focused on intracellular and cell wall polysaccharides [56]. Kashif et al. [57] showed that a treatment with 1-M NaOH in the biomass increased the yield and quality of polysaccharides of *Tetraselmis* sp. Dogra et al. [56] reported that the most efficient extraction of *T. suecica* polysaccharides was in the biomass treated with the Fenton reaction. This reaction increases the productivity of polysaccharides, because it generates oxidative stress to the microalgae biomass. Guzman-Murillo and Ascencio [26] extracted acid EPS from *T. suecica* and *Tetraselmis* sp. in which their maximum concentration was 409 mg L^−1^ and 1819 mg L^−1^, respectively, values higher than those obtained in our study. The difference between our results and the ones previously cited [26] could be due to the different salinities used in the culture media. In our study, we used a salinity of 35 ‰, while the aforementioned authors used salinities of 3–6 ‰, which would indicate that *T. suecica* produces a greater amount of acidic EPS at low salinities. The osmotic adjustment and the regulation of the turgor pressure of the microalgae is affected by salinity, since, when it is low, the cellular ionic concentrations increase and their ionic relationships are constant. On the contrary, at salinities greater than 20‰, the ionic relationships are variable [58]. It is important to consider that these variations in ionic relationships play a fundamental role in the excretion of polysaccharides. Furthermore, these variations will also depend on the species and its adjustment mechanisms to osmotic stress. For example, in the case of *Botryococcus braunii*, the increase in salinity allowed a greater production of polysaccharides [59]. Therefore, the increase in EPS production in *T. suecica* will depend on the cultivation condition, abiotic factors such as salinity and optimization of EPS extraction methods.

### 2.7. Elemental Analysis of Exopolysaccharides (EPS) of Autotrophic and Heterotrophic Biomass Cultures of T. suecica

The acid EPS obtained from the heterotrophic culture of *T. suecica* showed the highest content of carbon and nitrogen (*p* < 0.05; Table 5), while acid and total EPS obtained from the autotrophic culture of *T. suecica* presented the lowest content of carbon and nitrogen, respectively. Although no sulfur was found in the autotrophic EPS, this element was present in the heterotrophic EPS, being statistically higher in acid EPS (*p* < 0.05). Total autotrophic EPS had the highest C/N ratio (*p* < 0.05), while the lowest C/N ratio was found in acid autotrophic EPS (*p* < 0.05). The EPS C/N ratio of microalgae, including *T. suecica*, has been poorly studied until now; therefore, the study of these relationships should be increased and specified. The sulfur content was only detected in the EPS of heterotrophic cultures. Within these, the acid EPS have a higher sulfur content than the total EPS (*p* < 0.05), mainly due to the fact that the extraction method is aimed exclusively at sulfated EPS.

### 2.8. Antioxidant Activity of Exopolysaccharides (EPS) of Autotrophic and Heterotrophic Biomass Cultures of T. suecica

The total and acid heterotrophic EPS were 1.8 and 2.2 times higher than the autotrophic ones, respectively (*p* < 0.05; Figure 3).

The EPS from marine microalgae have shown the ability to protect oxidative stress, avoiding the accumulation of free radicals and reactive oxygen species (ROS) [28]. Dogra et al. [56] and Kashif et al. [57] reported the reducing capacity to eliminate radicals generated by ABTS, DPPH and FRAP methods for total EPS in the autotrophy of *Tetraselmis* sp. However, in the present study, it was only measured by the ABTS method, in which the total autotrophic EPS presented similar results. In the case of acid autotrophic EPS, and acid and total heterotrophic EPS, to the best of our knowledge, there are no previous references to this study, this being the first report of antioxidant activity for these types of *T. suecica* exopolysaccharides.

The antioxidant activity could be related to the percentage of galacturonic and glucuronic acids present in the constitution of the EPS of *T. suecica* (see Section 2.9). The heterotrophic EPS of *T. suecica* had sulfate in their constitution (Table 6), and, according to Mendiola et al. [30] and Sun et al. [60], the content of uronic acids and sulfate are related to an increase in the reducing capacity of free radicals [28]. The increase of these elements in the constitution of the heterotrophic EPS with respect to the autotrophic ones of *T. suecica* contributed to the increase of antioxidant activity. Possible phenols from ESPs were removed by precipitation with polyvinylpyrrolidone. It should be noted that it was not measured by the DPPH method, because its extraction is carried out in organic solvent, so the EPS immediately precipitated.

### 2.9. Fourier-Transform Infrared Spectroscopy (FTIR) of Exopolysaccharides (EPS) of Autotrophic and Heterotrophic Biomass Cultures of T. suecica

FTIR spectroscopy of exopolysaccharides obtained from autotrophic and heterotrophic cultures of T. suecica showed the presence of various functional groups in all samples, such as hydroxyl or carbonyls groups (Figure 4). Although the autotrophic EPS of *T. suecica* did not present the sulfate group peak (Figure 4A,C), the heterotrophic ones did (Figure 4B,D). To the best of our knowledge, this is the first characterization of total and acid EPS extracted from autotrophic and heterotrophic cultures of *T. suecica*.

Different absorbance peaks were observed in the spectra of the autotrophic and heterotrophic EPS, indicating their functional groups (Figure 4). The strongest and widest signals were located between 3000 and 3500 cm^−1^, which were attributed to the vibration of the -OH and -NH2 groups, followed by -CH2-methyl residues between 2800–2950 cm^−1^ characteristic of polysaccharides [61,62,63]. The peaks located between 1500 and 1700 cm^−1^ were due to the vibrations of the C=O groups and the stretching of C-N and the bending of NH [56,62,63]. The peak corresponding to the sulfate groups (S=O) was found between 1370–1240 cm^−1^ [64,65], which is characteristic of sulfated polysaccharides in marine microalgae [66]. The polysaccharides presented in the fingerprint zone from 1400 cm^−1^ to 700 cm^−1^, presenting various stretching and deformations corresponding to the polysaccharides bonds (C-O-C, C-O-P, C-N and P=O) [67].

Dogra et al. [56] and Kashif et al. [57] carried out the FTIR analysis to the soluble fraction of polysaccharides of *Tetraselmis* sp. biomass in which they found the peak of 1650 cm^−1^ of vibrations of C=O in accordance with the present investigation. Furthermore, they found peaks between 1068–1079 cm^−1^ indicating -COOH with α helix amino acids (low molecular weight proteins) and 1049 cm^−1^ attributed to an aliphatic group with a possible increase in antioxidant activity. These two peaks were not shown in the present work. According to Meng et al. [68], the FTIR method is a validated spectroscopic method, which characterizes algae polysaccharides, determines variations of other primary metabolites and evaluates the physiology of microalgae. This characterization method could be used to temporarily observe the EPS excretion dynamics, and the comparison of the spectra with digital libraries would allow the identification of fractions of biotechnological interest and industrial application.

### 2.10. Gas Chromatography—Mass Spectrometry (GC-MS) of Exopolysaccharides (EPS) of Autotrophic and Heterotrophic Biomass Cultures of T. suecica

In the GC-MS spectrum of total EPS extracted from the autotrophic culture of *T. suecica*, the highest peak corresponds to galactopyranoside with a retention time of 27.37 min, followed by glucose, galactose and glucuronic acid (Appendix A). Other minor monosaccharides (mannose, arabinose and ribose) were identified.

In the GC-MS spectrum of acid EPS extracted from the autotrophic culture of *T. suecica*, the highest peak corresponds to glucose with a retention time of 28.93 min, followed by galactopyranoside, glucuronic acid and galactose (Appendix A). Other minor monosaccharides were identified as xylose, galacturonic acid, mannose and ribose.

In the GC-MS spectrum of total EPS extracted from the heterotrophic culture of *T. suecica*, the highest peak corresponds to mannose with a retention time of 26.05 min, followed by glucose, glucuronic acid and rhamnose (Appendix A). Other minor monosaccharides (galactose, galacturonic acid, ribose, and fucose) were identified.

In the GC-MS spectrum of acidic EPS extracted from the heterotrophic culture of *T. suecica*, the highest peak corresponds to mannose with a retention time of 26.06 min, followed by glucose, glucuronic acid and galactopyranoside (Appendix A). Other minor monosaccharides were identified as galacturonic acid, galactose, fucose, ribose and xylose.

According to our revision, to the best of our knowledge, the monosaccharides of the EPS of *T. suecica* have not been previously characterized. However, intracellular and cell wall polysaccharides of *T. suecica* have been characterized as having 3-deoxy-d-manno-oct-2-ulosonic acid (Kdo) (54%), 3-deoxy-lyxo-2-heptulosaric acid (Dha) (17%), galacturonic acid (21%) and galactose (6%) by GC-MS [69]. For *Tetraselmis striata*, similar cell wall monosaccharides were described by NMR spectroscopy [70,71,72]. Dogra et al. [56] carried out the high-performance anion exchange chromatography with a pulsed amperometric detector (HPAEC-PAD) analysis to the soluble fraction of polysaccharides of *Tetraselmis* sp. biomass, in which they found that the KCTC 12432 BP strain contained a higher percentage of galactose and glucose in a molar ratio (11.1:8.2) and the strain KCTC 12236 BP showed the peaks of rhamnose, galactose, glucose, mannose and xylose. Therefore, the constitution of the EPS of *T. suecica* is different from the intracellular and cell wall polysaccharides. However, they have a similar monosaccharide composition with the soluble fraction of polysaccharides from *Tetraselmis* sp.

The EPS from autotrophic and heterotrophic cultures of *T. suecica* are composed in a higher percentage of glucose (23–37%), glucuronic acid (20–25%), mannose (2–36%), galactose (3–25%) and galactoryranoside (5–27%) and in lower percentages of galacturonic acid (0.1–3%); arabinose (5%); xylose (0.3–3%) and ribose, rhamnose and fucose (1%) (Table 6). Differences were found between the monosaccharide constitution of the EPS of autotrophic and heterotrophic cultures of this study. The highest percentage of mannose and fucose were the heterotrophic EPS, while the highest amounts of galactose and glucose were detected in autotrophic EPS. Xylose was present only in acid EPS from both culture conditions, while arabinose and rhamnose were found only in the total autotrophic and heterotrophic EPS.

According to Xiao and Zheng [73], the variations in EPS percentage are due to a nutritional stress of the culture conditions of origin of each one of them. These authors indicated that the differences and changes at the physiological level of the microalgae caused by different culture conditions make the microalgae adapt and biosynthesize polysaccharides according to the environmental conditions. Despite the percentages of monosaccharides presented for each EPS, the particular weight of each of the monosaccharides must be analyzed, representing the same high values of uronic acids in heterotrophy (galacturonic and glucuronic acids). According to de Jesus Raposo [28], polysaccharides with high contents of uronic acids present high bioactivity.

### 2.11. Cytotoxic Effects on Tumor Cells of Exopolysaccharides (EPS) of Autotrophic and Heterotrophic Biomass Cultures of T. suecica

The results obtained in this study showed that the EPS obtained from *T. suecica* in autotrophic and heterotrophic cultures have high cytotoxic effects on tumor cells (Figure 5). In the human leukemia cell line HL-60, inhibitory concentration (IC_50_) of 36 µg mL^−1^ and 68 µg mL^−1^ were determined for acidic autotrophic and heterotrophic EPS, respectively, and IC_50_ of 1784 µg mL^−1^ and 5183 µg mL^−1^ for total autotrophic and heterotrophic EPS, respectively (Figure 5A). In the breast cancer cell line (MCF-7), the acidic autotrophic and heterotrophic EPS showed IC_50_ of 60 µg mL^−1^ and 141 µg mL^−1^, respectively, while the total autotrophic and heterotrophic EPS showed lower effects with IC_50_ of 9461 µg mL^−1^ and 9135 µg mL^−1^, respectively (Figure 5B). In the case of the lung cancer cell line (NCI-H460), the acidic autotrophic and heterotrophic EPS showed high cytotoxic effects with IC_50_ of 118 µg mL^−1^ and 110 µg mL^−1^, respectively, whilst the total autotrophic and heterotrophic EPS had lower activity with IC_50_ of 5160 µg mL^−1^ and 8000 µg mL^−1^, respectively (Figure 5C).

This is the first evidence that EPS from *T. suecica* have cytotoxic effects on tumor cells. Previously, it has only been demonstrated that acidic EPS from *Tetraselmis* sp. inhibited the adhesion of *Helicobacter pylori* to HeLa S3 cells, indicating a possible prophylactic treatment in microbial infections, although in vivo experimental models are necessary [26]. Microalgae polysaccharides are interesting candidates for antitumor therapies. Polysaccharides from *Tribonema* sp. and *Phaedactylum tricornum* induced apoptosis in the liver cancer cell line (HepG2) [74,75]. Polysaccharides from *Artrosphira platensis* reduced cell proliferation in HepG2 and the breast cancer cell line (MCF-7) [76]. Other studies have described the antiproliferative activity of EPS from *Porphyridium cruentum* in the human cervical cancer cell line (HeLa) [77], MCF-7 cell line [78] and the inhibition of tumor growth Graffi myeloids [32]. Therefore, EPS from marine microalgae can be used as functional ingredients in foods or possible nutraceuticals to decrease the likelihood of tumor formation and development in the human body.

### 2.12. Cytotoxic of Exopolysaccharides (EPS) of Autotrophic and Heterotrophic Biomass Cultures of T. suecica

The autotrophic and heterotrophic total EPS did not reach the IC_50_ at the concentrations tested; therefore, they did not have a cytotoxic effect on the proliferation of the gingival fibroblast cell line (HGF-1) (Figure 6). However, the autotrophic and heterotrophic acid EPS showed cytotoxicity effects with IC_50_ of 165 µg mL^−1^ and 61 µg mL^- 1^, respectively (Figure 6). The elemental characteristic of cancer chemotherapeutics is that the compounds used do not affect the normal cell growth and have specific cytotoxicity [79]. Gingival fibroblast cell line (HGF-1) is a representative mammalian cell line that has been used for the investigation of anticancer activity [80]. Based on our results, the acids EPS showed high cytotoxicity; therefore, they are not suitable for therapeutic use. In contrast, the total EPS did not show cytotoxicity, so they could be a good candidate for anticancer investigation. However, it is important to deepen the studies and test other types of healthy cell lines.

## 3. Materials and Methods

### 3.1. Biological Material

The strain used in this experiment, *T. suecica* (Chlorophyta), is part of the microalgae bank collection (strain code nº UMA-260920) of the Institute of Biotechnology and Blue Development (IBYDA) of Malaga University (Malaga, Andalucia, Spain).

### 3.2. Culture Conditions

#### 3.2.1. Autotrophic Culture of *T. suecica*

The microalgae *T. suecica* was cultured in 100-mL glass photobioreactors with a diameter of 3.0 cm, with constant temperature (21 °C). The photoperiod was adjusted to 12 h of light (irradiance 165 µmol photons m^-2^ s^−1^) and 12h of darkness (12:12) and was maintained in agitation with a constant air bubbling system in order to avoid microalgae sedimentation and achieving a homogeneous distribution of nutrients and irradiance in each cell. The culture was inoculated with a concentration of 6 × 10^5^ cells mL^−1^ and adjusted to 35‰ salinity. The culture system was through a batch culture, with the addition of the culture medium F/2 [81]. The culture was volumetrically scaled to 10 L and was kept in a stationary phase. All the experiments were carried out in triplicate.

#### 3.2.2. Heterotrophic Culture of *T. suecica*

The adaptation of an autotrophic to a heterotrophic culture of *T. suecica* was carried out in 100-mL glass photobioreactors with a diameter of 3.0 cm and constant temperature (21 °C). A batch culture system was started. The reduction of the photoperiod illumination time was progressive (light:dark) 12:12 (288h), 8:16 (228h), 4:20 (240h), 3:21 (156h), 2:22 (168h) and 0:24. The culture medium used was F/2 supplemented with penicillin/streptomycin 0.1% *v/v*, with an optimal glucose dosage of 5 g L^−1^ in darkness condition. The culture was maintained with constant agitation by means of an air bubbling system, avoiding the sedimentation of the microalgae and achieving a homogeneous distribution of nutrients. Once the cultures were adapted to heterotrophy, they were volumetrically scaled to reach 2 L. All the experiments were carried out in triplicate.

### 3.3. Extraction Conditions

#### 3.3.1. Biomass Extraction

In autotrophic and heterotrophic conditions, when the stationary phase was achieved, the culture was harvested by centrifugation at 3500 rpm for 8 min at 4 °C. The biomass was washed twice with distilled water to eliminate salts traces and was subsequently dried in an oven at 40 °C for 48 h. Once dried, it was stored at room temperature until further analyses.

#### 3.3.2. Exopolysaccharides Extraction

Once the autotrophic and heterotrophic cultures reached the stationary phase, they were centrifuged at 3500 rpm for 8 min at 4 °C, and the supernatant was used for the extraction of total and acid EPS from *T. suecica*. Before extraction, phenols were removed by precipitation with polyvinylpyrrolidone (Sigma-Aldrich, St. Louis, MO, USA) and centrifugation 4500 rpm, 5 min, 4 °C. For this, total EPS were precipitated with the addition of ethanol (*v/v*) [82] and acid EPS with N-cetylpyridinium bromide (Cetavlon) (Sigma-Aldrich, St. Louis, MO, USA) 2% (*w/v*) for 24h [83]. Then, the EPS were centrifuged at 4500 rpm for 5 min, 4 °C. The supernatant was discarded, and 10 mL of 4-M NaCl (Sigma-Aldrich, St. Louis, MO, USA) was added. This mix was stirred until completely dissolved. Once cooled, ethanol was placed in a ratio (*v/v*) and kept at 4 °C for 24 h. After centrifugation at 4500 rpm, 5 min, 4 °C, the pellet containing the polysaccharides and salts was placed on a dialysis membrane (Sigma-Aldrich, St. Louis, MO, USA) in a 0.5-M NaCl solution overnight at 4 °C. Then, the dialyzed EPS was centrifuged at 4500 rpm for 5 min, 4 °C, and washed with absolute ethanol. Finally, acid and total EPS from both culture conditions (per triplicate, *n* = 3) were stored at −80 °C and subsequently freeze-dried at −50 °C. The EPS were quantified on an analytical balance, and the maximum concentrations obtained in the autotrophic and heterotrophic cultures were compared according to the culture volume and cell density.

### 3.4. Population Parameters (Cell Density, Cell Concentration, Specific Growth Rate, Biovolume and Cell Volume)

Cell concentration was calculated according to Equation 1 [84]:(1)$$Cell\;concentration=\;\left(\frac{W_{1}-W_{0}}{Volume}\right)$$ where W_1_ and W_0_ are the differences in total weight of cells g L^−1^ at volume filtration.

Through cell quantification in a 100-µm cuvette (Beckman Coulter™ AccuComp Z2, Indianapolis, IN, USA), at a pre-established dilution of 1/20, the specific growth rate μ (day^−1^) and the daily doubling of biomass was calculated, and the results were shown as the sum of these with the algorithm of Arredondo and Voltolina [84], summarized in Equation (2):(2)$$µ=\frac{\ln\left(N_{2}-N_{1}\right)}{t_{2}-t_{1}}$$ where *N*_2_ and *N*_1_ are the density of cells mL^−1^ at times *t*_2_ and *t*_1_, respectively.

The biovolume was calculated from the mean of the population (µm^3^ cell^−1^) provided by cell quantification and the value of the cell density (cell mL^−1^) to obtain its final value in µm^3^ mL^−1^. 

### 3.5. Total Carbon (C), Hydrogen (H), Nitrogen (N) and Sulfur (S) of Dry Biomass and Exopolysaccharides

Total carbon (C), nitrogen (N) and sulfur (S) were determined from dry biomass and the extracted polysaccharides using the total combustion technique used in the LECO TruSppec Micro CHNSO-Elemental Analyzer (St. Joseph, MI, USA). This technique is based on the complete and instantaneous oxidation of the sample by pure combustion with controlled oxygen at a temperature of up to 1050 °C (C, H, N and S) and pyrolysis at 1300 °C (O) for decomposition of O as CO and oxidation to CO_2_. The resulting combustion products, CO_2_, H_2_O, SO_2_ and N_2_ are subsequently quantified by selective IR Pleabsorption detector (C, H and S) and TCD (N) differential thermoconductivity sensor. The result of each element (C, H, N and S) is expressed in % with respect to the weight of the sample.

### 3.6. Biochemical Composition of Autotrophic and Heterotrophic Biomass Cultures of T. suecica

For the biochemical composition (*n* = 3), total proteins were calculated from the elemental N determination using the N-protein conversion factor of 4.80 reported by Lourenço et al. [85]; lipids were extracted according to the Folch method [86] and carbohydrates according to the phenol-sulfuric procedure [87]. Moisture and ash levels were determined gravimetrically by drying in an oven at 105 °C and after incineration in a muffle furnace at 550 °C, respectively, until constant weight.

### 3.7. Determination of Phenolic Compounds

Quantification of phenolic compounds was performed according to the Folin-Ciocalteu method [88]. Reaction was performed by adding 20 mg of biomass and placed in an Eppendorf with 1 mL of 80% methanol (Sigma-Aldrich, St. Louis, MO, USA). The solution was stirred and incubated at 4 °C in darkness for 12 to 24h. Following this, the solution was centrifuged at 3500 rpm for 10 min, 4 °C. Then, 100 μL of the extract, 700 μL of distilled water, 150 μL of 20% anhydrous sodium carbonate (Na_2_CO_3_) (Sigma-Aldrich, St. Louis, MO, USA) and 50 μL of the Folin-Ciocalteu reagent (Sigma-Aldrich, St. Louis, MO, USA) were mixed by stirring and incubated at 4 °C in darkness for 2h. The absorbance was measured at 760 nm using a UV–visible spectrophotometer (SHIMADZU UV MINI-1240, Duisburg, Germany). The blank included all reagents, except the extract that was replaced by 80% methanol. Phenolic contents were determined by constructing a standard curve using different phloroglucinol (Sigma-Aldrich, St. Louis, MO, USA) concentrations. Results were expressed in the mg equivalent of phloroglucinol per g of algal dry weight (DW).

### 3.8. Determination of Pigments

For chlorophyll a, chlorophyll b and total carotenoid quantification, an acetone 90% extract was done with 5 mg of freeze-dried biomass in 1 mL of solvent. The extract was sonicated for 3 min and remained 24 h in darkness at 4 °C. Chlorophyll a was determined according to Equation (3) [89], chlorophyll b was determined according to Equation (4) [90] and total carotenoids were determined according to Equation 5 [91].
[(11.8668 × A_664_) − (−1.7858 × A_647_)](3)
[(18.9775 × A_647_) − (−4.8950 × A_664_)](4)
[(A_480_ × 4.0)] (5)
where A_664_, A_647_ and A_480_ are the measured absorbance at 667, 647 and 480 nm, respectively.

### 3.9. Lipopolysaccharides (LPS) Contamination Assay

The presence of lipopolysaccharides (LPS) in the EPS fractions isolated from *T. suecica* was evaluated using the *Limulus* amebocyte lysate (LAL) assay kit (Endosafe^®^-PTS, Charles River Laboratories, Charleston, SC, USA). In brief, 25 μL of the EPS solution (concentration 50 μg mL^−1^) in distilled water was loaded into each of the four channels of the cartridge. The reader automatically mixed the sample with the LAL reagent in two channels. Additionally, the LAL reagent was mixed with the positive control in the other two channels. These samples were used as the control. Afterward, all samples were incubated and combined with the chromogenic substrate. After mixing, the optical density of the four channels was measured and compared with an internal standard curve. The amount of endotoxin in the sample was expressed as endotoxin units (EU) mL^−1^.

### 3.10. Antioxidant Capacity

#### 3.10.1. ABTS Assay Scavenging of Free Radical in Exopolysaccharides and Biomass

The ability of the polysaccharides to scavenge the free radicals was evaluated using an ABTS assay according to Re et al. [92], with few modifications. ABTS radical cation was produced through the reaction with the 2,2’-azino-bis (3-ethylbenzothiazoline-6-sulphonic acid) (ABTS) 7-mM solution with 2.45-mM potassium persulfate for 16h in the dark at 4 °C. After incubation, the well-mixed solution was diluted to an absorbance of 0.7 at 727 nm with the deionized water. For biomass and EPS, 10 mg were weighted, and 1 mL of phosphate buffer was added, mechanically disrupted and centrifuged at 4 °C for 5 min at 3500 rpm. A total of 50 μL of these samples (supernatant in case of biomass) were mixed with 940 μL of phosphate buffer and 10 μL of ABTS solution. The resulting mixture was measured with a spectrophotometer at 727 nm. ABTS radical scavenging capacity was calculated according to Equation (6) [93]:AA% = (Abs_0_ − Abs_1_ /Abs_0_) × 100 (6)
where Abs_0_ is the absorbance of the ABTS radical in phosphate buffer at time 0, and Abs_1_ is the absorbance of the ABTS radical solution mixed with the sample after 8 min. A calibration curve was performed with different concentrations of Trolox^®^ (0 to 5 μg mL^−1^) from a stock of Trolox^®^ 2.5 mM. The % inhibition was determined by interpolation of the absorbance values in the Trolox standard curve fitted to the equation of a linear regression line (y = 13.593x + 0.8717; R^2^ = 0.99). All determinations were performed in triplicate (*n* = 3).

#### 3.10.2. DPPH Free-Radical Method in Biomass from *T. suecica*

Radical scavenging and antioxidant activities of the extracts of biomass were assessed by the 2,2-diphenyl-1-picrylhydrazyl (DPPH) free-radical method by Brand-Williams et al. [94]. To obtain the extracts from the biomass, 10 mg of the sample was weighted, and 1 mL of 80% methanol was added, homogenized by mechanical disruption, and left for 16 h in the dark at 4 °C. After incubation, samples were centrifuged at 4 °C for 5 min at 3500 rpm. Aliquots of 200 μL of the supernatant samples were added to 90 µL of instantly prepared DPPH solution (0.358 mM) and 910 µL 80% methanol. The samples were incubated in the dark for 30 min at room temperature, and the absorbance (abs) was read at 517 nm against 80% methanol as blank. The absorbance was then transformed into a percentage of inhibition versus 80% methanol. The percentage of the antioxidant activity was calculated according to Equation (7) [93].
AA% = [(Abs_0_ − Abs_1_) Abs_0_] × 100(7)
where Abs_0_ is absorbance at time zero, and Abs_1_ is absorbance at the end of the reaction (30 min) at 517 nm. A calibration curve was performed with different concentrations of Trolox^®^ (0 to 7.5 μg mL^−1^) from a stock of Trolox^®^ 2.5 mM. The % inhibition was determined by interpolation of the absorbance values in the Trolox standard curve fitted to the equation of a linear regression line (y = 13.593x + 0.8717; R^2^ = 0.99). All determinations were performed in triplicate (*n* = 3).

### 3.11. Fourier-Transform Infrared Spectroscopy (FTIR)

Fourier-transform infrared (FTIR) spectra of the polysaccharides from *T. suecica* were obtained by using self-supporting pressed discs of 13 mm in diameter of a mixture of polysaccharides and KBr (1% *w/w*) with a hydrostatic press at a force of 15.0 tcm^−2^ for 2 min. The FTIR spectra were obtained with a Thermo Nicolet Avatar 360 IR spectrophotometer (Thermo Electron Inc., Franklin, MA, USA) having a resolution of 4 cm^−1^ with a deuterated triglycine sulfate (DTGS) detector and using Omnic^TM^ 7.2 software (bandwidth 50 cm^−1^ and enhancement factor 2.6) in the 400–4000 cm^−1^ region. Baseline adjustment was performed using the Thermo Nicolet OMNIC software to flatten the baseline of each spectrum. The OMNIC correlation algorithm was used to compare sample spectra with those of the spectral library (Thermo Fisher Scientific, San Jose, CA, USA).

### 3.12. Gas chromatography–Mass Spectrometry (GC-MS)

#### 3.12.1. Hydrolysis and Derivatization of EPS

Polysaccharides samples (2 mg) and monosaccharides standards were treated with the same procedure. First, 100 µL of the standard stock solution of 1 mg mL^−1^ of each monosaccharide was dried under nitrogen gas flow. Second, the samples of polysaccharides, and a mixture containing the standard monosaccharides included in the IS (Internal Standard), were methanolized in 2-mL methanol/3-M HCl at 80 °C during 24 h. The monosaccharides glucose, galactose, rhamnose, fructose, mannose, xylose, apiose and myo-inositol (Internal Standard, IS), as well as pyridine, hexane and methanol/3-M HCl solution, were purchased from Sigma-Aldrich. Then, the saccharides were washed with methanol and dried under nitrogen gas flow. Third, the trimethylsilyl reaction was accomplished with 200 µL of Tri-Sil HTP (Thermo Fisher Scientific, Franklin, MA, USA). Each vial with the sample was heated at 80 °C for 1 h. The derivatized sample was cooled to a room temperature and dried under a steam of nitrogen. Forth, the dry residue was extracted with hexane (2 mL) and centrifuged. Finally, the hexane solution containing silylated monosaccharides was concentrated and reconstituted in hexane (200 µL), filtered and transferred to a GC-MS autosampler vial. Sample preparation and analyses were performed in triplicate.

#### 3.12.2. Gas Chromatography/Mass Spectrometry (GC-MS) Analysis

GC/MS analyses were carried out using a gas chromatograph Trace GC (Thermo Fisher Scientific, Franklin, MA, USA), an autosampler Triplus RSH (Thermo Fisher Scientific, Franklin, MA, USA) and a DSQ mass spectrometer quadrupole (Thermo Fisher Scientific, Franklin, MA, USA). The GC column was set ZB-5 Zebron, Phenomenex (5% phenyl and 95% dimethylpolysiloxane), 30-m (length) × 0.25-mm (I.D) × 0.25-μm film thickness. Injection volume was 1 μL in split mode, with a split ratio of 40. Helium was used as the carrier gas with a flow rate of 1.2 mL min^−1^. The injector was set at 250 °C in split mode. The initial oven temperature was 80 °C for 2 min, then ramped from 10 °C min^−1^ to 180 °C, followed by ramping from 5 °C min^−1^ to 250 °C, remaining constant for 2 min. The electron impact ionization (EI) mode of the mass spectrometer was set at 70 eV. Monitored in full scan mode with mass range 50-650 *m/z* with interface temperature 250 °C and ionization source temperature of 230 °C. The identification of monosaccharides in polysaccharide samples was carried out by comparing retention time and mass spectra of monosaccharide standards, previously analyzed under identical conditions (glucose, galactose, mannose, arabinose, xylose, rhamnose, ribose, fucose, galacturonic acid and glucuronic acid). The compounds were identified by comparing the mass spectra with those in the National Institute of Standards and Technology (NIST 2014) library.

### 3.13. Cell Line Cultures

In this study, four cell lines were used: three of them tumoral, such as the human breast adenocarcinoma cell line (MCF-7, ATCC, Manassas, VA, USA), human leukemia cell line (HL-60 ATCC), human lung cancer cell line (NCI-H460, ATCC) and an immortalized human gingival fibroblast-1 cell line HGF-1 (ATCC CRL-2014). MCF-7 and HGF-1 cell lines were routinely cultured in Dulbecco’s modified Eagle’s medium (DMEM) (Biowest, Barcelona, Spain) supplemented with 10% fetal bovine serum (Biowest, Barcelona, Spain), 1% penicillin-streptomycin solution 100X and 0.5% of amphotericin B (Biowest, Barcelona, Spain), while NCI-H460 cells were cultured in Roswell Park Memorial Institute (RPMI-1640) medium (Biowest, Barcelona, Spain) supplemented with 10% fetal bovine serum, 1% penicillin-streptomycin solution 100X and 0.5% of amphotericin B, and HL-60 cells were cultured in RPMI-1640 medium (Biowest, Barcelona, Spain) supplemented with 20% fetal bovine serum, 1% penicillin-streptomycin solution 100X and 0.5% of amphotericin B (Biowest, Barcelona, Spain). Cells were maintained sub-confluent at 37 °C in humidified air containing 5% CO_2_. Cultured cells were collected by gentle scraping when confluence was reached 75% in the case of the MCF-7, HFG-1 and NCI-H460, as they are adherent cells. The scrapping of HL-60 cells was not performed, because these are cells in suspension. So, they were collected by centrifugation at 1500 rpm for 5 min.

### 3.14. Cytotoxic Effects on Tumor Cells Assay

For cytotoxic effects on the tumor cells assay, HL-60, MCF-7 and NCI-H460 cell lines were incubated at 2 × 10^4^, 8 × 10^3^ and 8 × 10^3^ cell lines per well, respectively in the presence of different concentrations (19 − 1 × 10^4^ µg mL^−1^) of EPS. The experiment was conducted individually with each cell line in a 96-well microplate (Sarstedt, Nümbrecht, Germany) for 72 h. The incubation conditions were as follows: temperature 37 °C, 5% CO_2_ and a humid atmosphere. As a control, the same cell lines were used without treatment. The proliferation of these cell lines was estimated by the MTT (Sigma-Aldrich, St. Louis, MO, USA) (3-(4,5-dimethylthiazol-2-yl)-2,5-diphenyltetrazolium bromide) assay [31]. Briefly, a volume of 10 μL of the MTT solution (5 mg mL^−1^ in phosphate-buffered saline) was added to each well. The microplates were incubated at 37 °C for 4 h. The yellow tetrazolium salt of MTT was reduced by mitochondrial dehydrogenases of metabolically active viable cells to form insoluble purple formazan crystals. Formazan was dissolved by the addition of sulfated–isopropanol (150 μL of 0.04-N HCl–2 propanol) and measured spectrophotometrically at 550 nm (Micro Plate Reader 2001, Whittaker Bioproducts, Promega, Wisconsin, USA). The relative cell viability was expressed as the mean percentage of viable cells compared with untreated cells. Four samples for each tested concentration were included in each experiment. Measurements were carried out in triplicate independent experiments.

### 3.15. Cytotoxicity Assay in Healthy Cell Line

For the cytotoxicity assay, HGF-1 cells were incubated at 1.5 × 10^4^ cells per wells in the presence of different concentration of EPS (19−1 × 10^4^ µg mL^−1^) in a 96-well microplate (Sarstedt, Nümbrecht, Germany) for 72 h. The incubation conditions were as follows: temperature 37 °C, 5% CO_2_ and a humid atmosphere. Cells proliferation was estimated by the MTT (Sigma-Aldrich, St. Louis, MO, USA) (3-(4,5-dimethylthiazol-2-yl)-2,5-diphenyltetrazolium bromide) assay [31], as explained above.

### 3.16. Statistical Analysis

For the statistical analysis of the experimental results, the STATISTICA software (V.7; Tulsa, OK, USA) was used. All values were expressed as mean ± standard deviations (SD). The one-way analysis of variance (ANOVA) was used to determine the differences between the EPS production, elemental analysis of EPS and antioxidant activity of EPS. When significant differences in the ANOVA were found, the post-hoc Tukey test was performed to identify the difference between the treatments. The Student’s *t*-test was used to determine significant differences between the cultures in all population parameters, biochemical composition, C/N ratio, phenolic compounds, antioxidant activity and pigments. Homogeneity of variance was checked using the Cochran test and by visual inspection of the residues [95]. The correlations of the data obtained were calculated using Pearson’s correlation analysis. Statistical significance of mean differences was considered to be attained with *p* < 0.05.

## 4. Conclusions

We determined that the heterotrophic conditions have several advantages over the autotrophic, such as improving the biochemical composition; enhancing the accumulation of proteins, lipids and carbohydrates and increasing the yield of exopolysaccharides (EPS). The antioxidant activity of the heterotrophic crop was higher with respect to the autotrophic crop, as well as for the algal biomass and EPS. In addition, to the best of our knowledge, this is the first time that autotrophic and heterotrophic EPS of *T. suecica* proved to have cytotoxic effects on HL-60, MCF-7 and NCI-H460 tumor cells. Therefore, they could offer greater benefits as possible natural nutraceuticals for the pharmaceutical industry. More studies are necessary to identify the specific bioactive fractions of each EPS.

## Figures and Tables

**Figure 1 marinedrugs-18-00534-f001:**
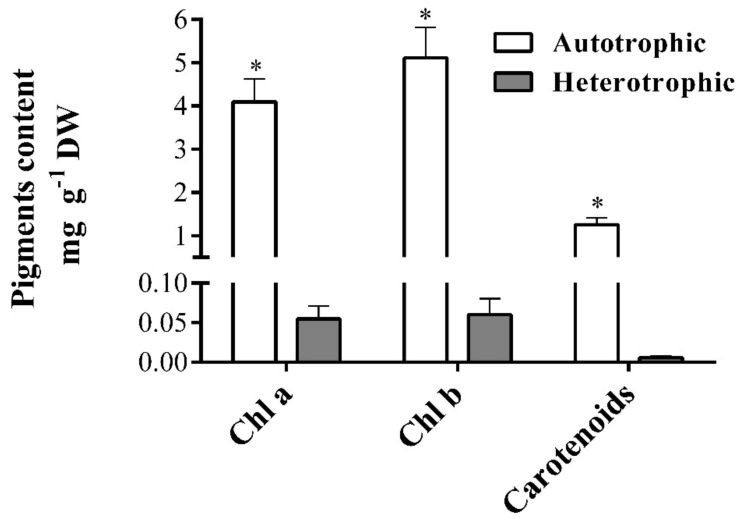
Pigment content extracted of the biomass from autotrophic and heterotrophic cultures of *Tetraselmis suecica* (mean ± SD; *n* = 3). Asterisks denote significant differences (*p <* 0.05, Student’s *t*-test). DW: dry weight, Chl a and Chl b: chlorophyll-a and chlorophyll-b.

**Figure 2 marinedrugs-18-00534-f002:**
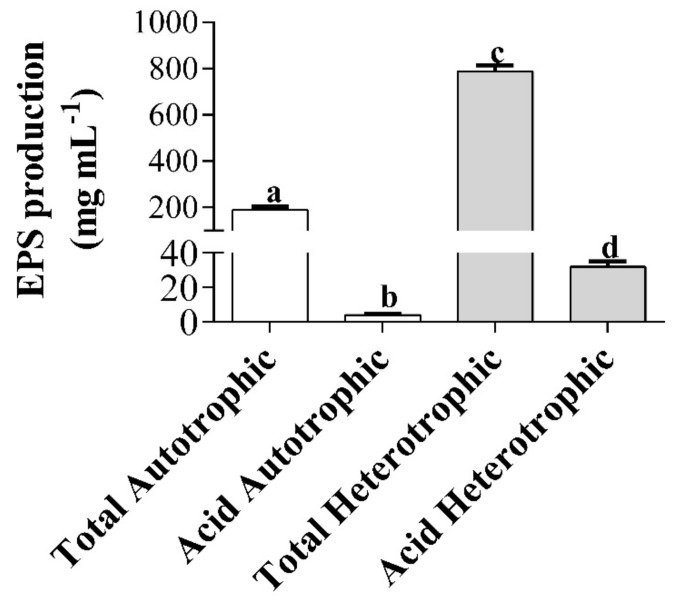
Total and acid exopolysaccharides (EPS) production from autotrophic and heterotrophic cultures of *T. suecica* (mean ± SD; *n* = 3). Different letters indicate significant differences (ANOVA, Tukey’s test, *p <* 0.05).

**Figure 3 marinedrugs-18-00534-f003:**
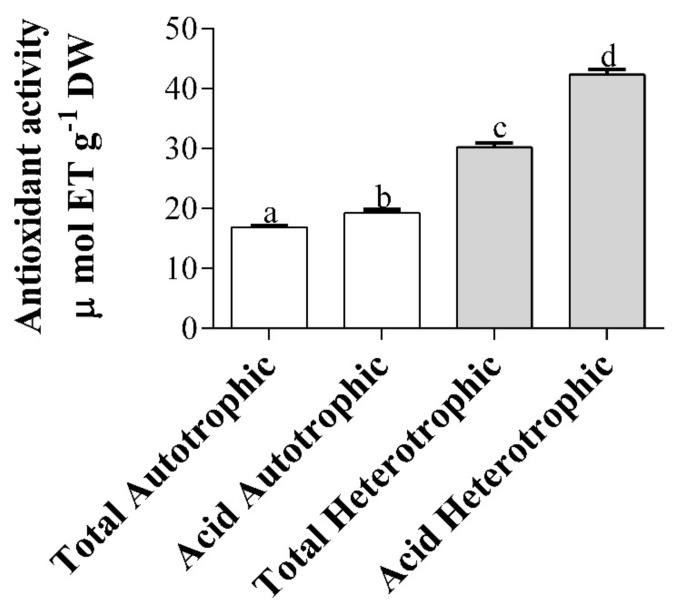
Antioxidant activity of total and acid exopolysaccharides from autotrophic and heterotrophic cultures of *T. suecica*. The antioxidant activity is expressed as micromoles of Trolox equivalents per gram of dry weight (µmol TE g ^– 1^ DW) (mean ± SD; *n* = 3). Different letters indicate significant differences (ANOVA, Tukey’s test, *p <* 0.05).

**Figure 4 marinedrugs-18-00534-f004:**
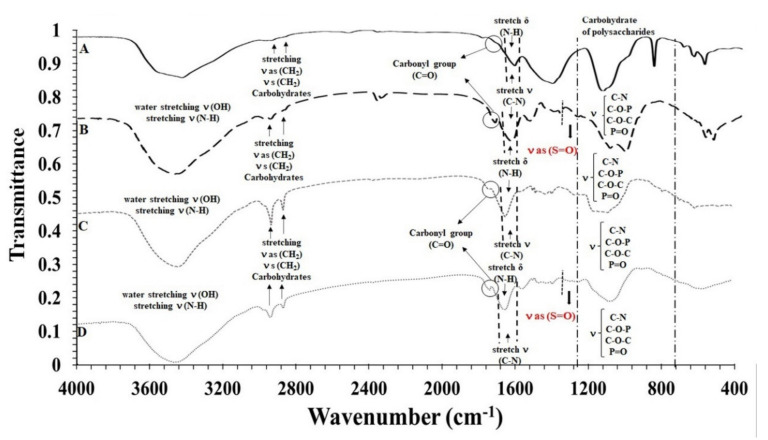
Fourier-transform infrared spectroscopy (FTIR) spectra of (**A**) total EPS obtained from the autotrophic culture of *T. suecica*, (**B**) total EPS obtained from the heterotrophic culture of *T. suecica*, (**C**) acid EPS obtained from the autotrophic culture of *T. suecica* and (**D**) acid EPS obtained from heterotrophic culture of *T. suecica.*

**Figure 5 marinedrugs-18-00534-f005:**
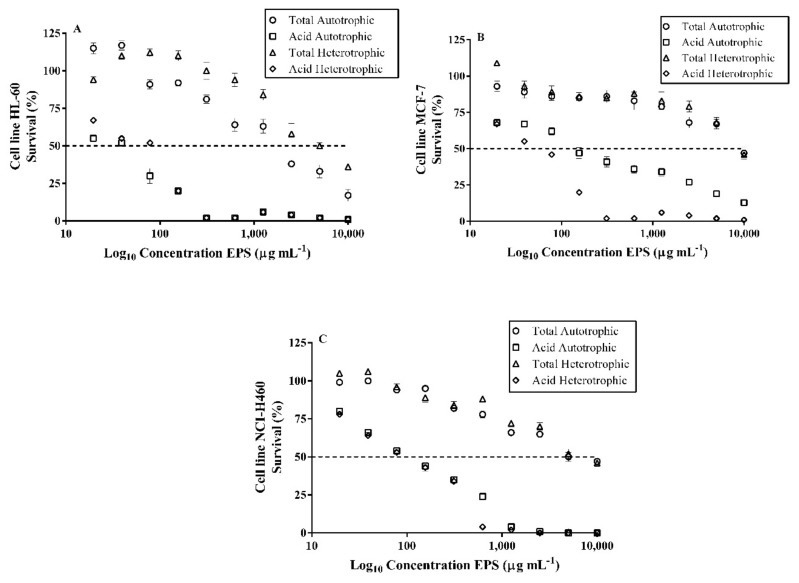
(**A**). Survival (%) of the human leukemia cell line (HL-60) exposed to different concentrations of EPS from *T. suecica*. (**B**). Survival (%) of the human breast cancer cell line (MCF-7) exposed to different concentrations of EPS from *T. suecica*. (**C**). Survival (%) of the human lung cancer cell line (NCI-H460) exposed to different concentrations of EPS from *T. suecica*.

**Figure 6 marinedrugs-18-00534-f006:**
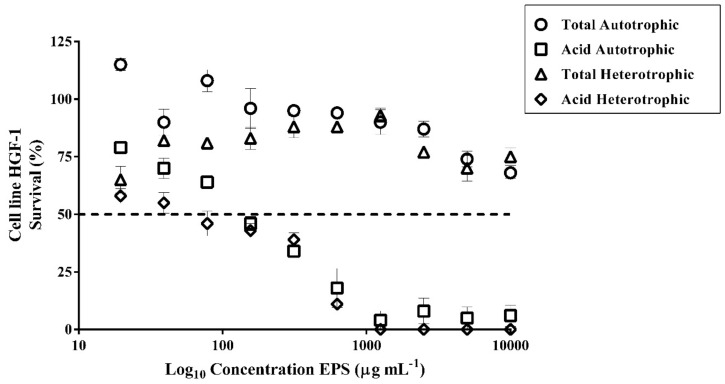
Survival (%) of the human gingival fibroblast cell line (HGF-1) exposed to different concentrations of EPS from *T. suecica***.**

**Table 1 marinedrugs-18-00534-t001:** Population parameter from autotrophic and heterotrophic cultures of *T. suecica* (mean ± SD; *n* = 3). Asterisks denote significant differences (*p* < 0.05, Student’s *t*-test).

Population Parameter	Autotrophic	Heterotrophic
Cell density (cell mL^−1^)	2.0 × 10^6^ ± 1.6 × 10^4^	3.6 × 10^7^ ± 7.7 × 10^4^ *
Cell concentration (g L^−1^)	5.5 ± 0.5	10.2 ± 0.7 *
Specific growth rate (µ) (d^−1^)	0.3 ± 0.2	0.3 ± 0.2
Biovolume (µm^3^ mL^−1^)	8.8 × 10^8^ ± 2.8 × 10^8^	1.2 × 10^9^ ± 3.0 × 10^8^ *
Cell volume (µm^3^)	521 ± 33 *	30 ± 2

**Table 2 marinedrugs-18-00534-t002:** Total carbon (TC), total nitrogen (TN) content and C/N ratio in the biomass obtained from autotrophic and heterotrophic cultures of *T. suecica* (mean ± SD; *n* =3). Asterisks denote significant differences (*p* < 0.05, Student’s *t*-test).

Elemental Analysis (%)	Autotrophic	Heterotrophic
TC	25.7 ± 0.4	31.0 ± 0.1 *
TN	3.5 ± 0.2	4.3 ± 0.1 *
C/N	7.3 ± 0.2	7.2 ± 0.1

**Table 3 marinedrugs-18-00534-t003:** Content of proteins, carbohydrates, lipids, ash and moisture in the biomass from autotrophic and heterotrophic cultures of *T. suecica* (% of dry weight (DW); mean ± SD; *n* = 3). Asterisks denote significant differences (*p* < 0.05, Student’s *t*-test).

Biochemical Composition	Autotrophic	Heterotrophic
Proteins	16.76 ± 0.40	20.78 ± 0.14 *
Lipids	6.13 ± 0.12	7.96 ± 0.10 *
Carbohydrates	24.31 ± 0.32	28.18 ± 0.37 *
Ash	34.88 ± 0.08	33.07 ± 1.30
Moisture	17.93 ± 0.52 *	10.01 ± 1.16

**Table 4 marinedrugs-18-00534-t004:** Total phenolic content and antioxidant capacity measured by 2,2’-azino-bis (3-ethylbenzothiazoline-6-sulphonic acid) (ABTS) and 2,2-diphenyl-1-picrylhydrazyl (DPPH) assay in the biomass from autotrophic and heterotrophic cultures of *T. suecica* (mean ± SD; *n* = 3). Asterisks denote significant differences (*p* < 0.05, Student’s *t*-test). TE: Trolox equivalents.

Culture	Phenols (mg Eq Phloroglucinol)	Antioxidant Activity	
		ABTS (µmol TE g^−1^ DW)	DPPH (µmol TE g^−1^ DW)
Autotrophic	3.88 ± 0.03	24.25 ± 0.70	3.49 ± 0.61
Heterotrophic	5.56 ± 0.10 *	80.17 ± 0.95 *	6.35 ± 0.91 *

**Table 5 marinedrugs-18-00534-t005:** Total carbon (TC), total nitrogen (TN), ratio C/N and sulfur (S) (%) obtained in the total and acid polysaccharides extracted from autotrophic and heterotrophic cultures of *T. suecica*. The data represent the average ± standard deviation (*n* = 3). Different letters indicate significant differences among polysaccharide types (ANOVA, Tukey’s test, *p* < 0.05).

Culture	Exopolysaccharide Type	% TC	% TN	C/N	% S
Autotrophic	Total	9.60 ± 0.10 ^c^	0.53 ± 0.01 ^d^	18.28 ± 0.17 ^a^	0.00
Autotrophic	Acid	9.02 ± 0.08 ^d^	0.71 ± 0.01 ^b^	12.75 ± 0.10 ^c^	0.00
Heterotrophic	Total	9.84 ± 0.07 ^b^	0.66 ± 0.02 ^c^	14.91 ± 0.10 ^b^	0.33 ± 0.08 ^b^
Heterotrophic	Acid	11.96 ± 0.09 ^a^	0.80 ± 0.01 ^a^	14.88 ± 0.12 ^b^	3.47 ± 0.10 ^a^

**Table 6 marinedrugs-18-00534-t006:** Percentage of principal monosaccharides obtained in the total and acid polysaccharides extracted from autotrophic and heterotrophic cultures of *T. suecica.*

Monosaccharide	Autotrophic Total (%)	Autotrophic Acid (%)	Heterotrophic Total (%)	Heterotrophic Acid (%)
Arabinose	5.23	-	-	-
Ribose	0.83	0.65	1.22	0.33
Rhamnose	-	-	1.36	-
Fucose	-	-	0.38	0.35
Xylose	-	3.03	-	0.30
Mannose	6.64	1.57	36.15	34.49
Galacturonic acid	<0.10	2.93	2.45	2.95
Galactopyranoside	5.11	27.06	6.76	8.14
Galactose	25.27	9.96	3.00	2.93
Glucose	35.46	34.70	23.32	37.57
Glucuronic acid	21.47	20.10	25.36	22.94

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
