# Peer review of "Antioxidant and Cytotoxic Effects on Tumor Cells of Exopolysaccharides from Tetraselmis suecica (Kylin) Butcher Grown Under Autotrophic and Heterotrophic Conditions"

_marinedrugs, 2020, doi:10.3390/md18110534_

Round 1

Reviewer 1 Report

In the manuscript entitled: "Antioxidant and antitumor activity of exopolysaccharides from Tetraselmis suecica grown under autotrophic and heterotrophic conditions" the Authors analyzed  antioxidant and anti-tumor activities of Tetraselmis suecica.

Authors should better clarify the results and discussions of 2.1 section (2.1. Adaptation of autotrophic to heterotrophic culture of T. suecica.

The authors talk about the antitumor activity of Tetraselmis suecica, perhaps it would be better to say cytotoxicity effect on tumor cells of T. suecica.

The Authors in the title and in the abstract must insert the binomic classification (Kylin (Butcher) of Tetraselmis sueica and in the whole manuscript write the abbreviated name (T. sueica) and in italics (page 2 line 59; page 5 line 175, 182 etc ..... )

Theobroma cacao (page 5 line 169), C. vulgaris, S. obliqus (page 5 line 171), Helicobcater pylori (page 13 line 408), Paheadactylum tricornum (page 13, line 410), Astrophira platensis (page 13 line 412), Porphyridum (pag. 13 line 413) and all algae names must be written in italics.

Author Response

Response to Reviewer 1 Comments

We thank the reviewer for their constructive criticism, and time spent to analyze this manuscript. The responses, and explanations related to their comments are listed below:

The review report of reviewer indicate that the conclusions must be improved.

The comments of the reviewer have been considered. The final conclusions have been changed in the manuscript (page 20 line 765).

“We have determined that the heterotrophic conditions have several advantages over the autotrophic, such as improving the biochemical composition, enhance the accumulation of proteins, lipids and carbohydrates and increases the yield exopolysaccharides (EPS). The antioxidant activity of the heterotrophic crop was higher with respect to the autotrophic crop, as well as for the algal biomass and EPS. In addition, to the best of our knowledge, this is the first time that autotrophic and heterotrophic EPS of T. suecica proved to have cytotoxic effects on HL-60, MCF-7 and NCI-H460 tumor cells. Therefore, they could offer greater benefits as possible natural nutraceuticals for the pharmaceutical industry. More studies are necessary to identify the specific bioactive fractions of each EPS.”

Point 1: Authors should better clarify the results and discussions of 2.1 section (2.1. Adaptation of autotrophic to heterotrophic culture of T. suecica.

We agree with the reviewer's comment on this issue. Section 2.1 of results and discussions have been reviewed, shortened and clarified as stated below (page 2 line 85):

“The heterotrophic culture of T. suecica showed that cell density, cell concentration and biovolume were statistically higher in heterotrophic culture (p < 0.05) while the cell volume was 17 times lower compared with the autotrophic culture (p < 0.05) (Table 1). The specific growth rate between autotrophic and heterotrophic culture no showed statistic differences significantly (Table 1).

Azma et al. [12] obtained differences in the final cell concentration of T. suecica grown in autotrophy and heterotrophy. On the contrary, Day and Tsavalos [33] found no differences in the final cell concentration of Tetraselmis sp. between the two culture conditions. These variations could be due to growth in the absence of light and the presence of organic substrates can change the metabolism and morphology of cells. In our investigation, glucose was used as the source of organic carbon, which generated high cellular concentrations due to the energy provided (2.8 kJ mol-1), compared to the 0.8 kJ mol-1 for acetate used in Azma et al. [12] investigation.

The adaptation of autotrophic to heterotrophic culture was performed by the progressive reduction of the illumination times in the photoperiod, preserving the irradiance of the T. suecica cultures. However, Azma et al. [21] made a progressive decrease in lighting for T. suecica cultures with longer periods, adding a total of 1650 hours compared to the present study, which was 1080 hours for adaptation to heterotrophy, meaning 35% less hours of adaptation, which would be due to the different media used cultivation. The Walne medium [34] used in Azma et al. [21] investigation contains concentrations of nitrate, phosphate, EDTA, zinc, molybdenum and manganese higher than F/2 used in the present study. Therefore, T. suecica has the ability to regulate its metabolism to achieve balanced growth in heterotrophic culture, this capability can be used to increase the production of metabolites of biotechnological interest.”

[12] Azma, M. et al. Biochem Eng J 2011, 53, 187–195.

[21] Azma, M. et al. Open Biotechnol J 2010, 4, 36–46.

[33] Day, J., Tsavalos, A. J Appl Phycol 1996, 8, 73–77.

[34] Walne, P. Fish Invest Lond Ser 1970, 1-62.

Point 2: The authors talk about the antitumor activity of Tetraselmis suecica, perhaps it would be better to say cytotoxicity effect on tumor cells of T. suecica.

Comments by the reviewer have been considered. We have changed the name "antitumor activity" to "cytotoxic effects on tumor cells" in order to improve the paper.

Point 3: The Authors in the title and in the abstract must insert the binomic classification (Kylin (Butcher) of Tetraselmis suecica and in the whole manuscript write the abbreviated name (T. suecica) and in italics (page 2 line 59; page 5 line 175, 182 etc ..... ).

As a response to the reviewer’s comment, we have inserted the binomic classification (Kylin (Butcher) of Tetraselmis suecica in the title and abstract. Furthermore, we have revised that throughout the text the abbreviated name "T. suecica" is written in italics.

Point 4: Theobroma cacao (page 5 line 169), C. vulgaris, S. obliqus (page 5 line 171), Helicobcater pylori (page 13 line 408), Paheadactylum tricornum (page 13, line 410), Astrophira platensis (page 13 line 412), Porphyridum (pag. 13 line 413) and all algae names must be written in italics.

As a response to the reviewer’s comment, we have written in italics all scientific names throughout the manuscript.

Reviewer 2 Report

The manuscript entitled "Antioxidant and antitumor activity of exopolysaccharides from Tetraselmis suecica grown under autotrophic and heterotrophic conditions" by Parra-Riofrio et al., present interesting data which deserves publication in Marine Drugs, however, the manuscript is too long, and it should be shortened. The text is clear and well written. References are abundant, actalized and well treated.

I propose that:

1. Figures 5, 6, 7, and 8 should go to supplementary data, because the relevant information supplied by them are harvested in Table 6 of the manuscript.

2. In fig. 3, I miss the data corresponding to the nucleic acids in different cells culture. These data are relevant for many biotechnological applications of microalgae.

Author Response

Response to Reviewer 2 Comments

We thank the reviewer for their constructive criticism, and time spent to analyze this manuscript. The responses, and explanations related to their comments are listed below:

Point 1: Figures 5, 6, 7, and 8 should go to supplementary data, because the relevant information supplied by them are harvested in Table 6 of the manuscript.

As a response to the reviewer’s comment, we have placed the Figures 5, 6, 7 and 8 as a Supplementary data. As a result, we have changed the name of the figure to: "Figure 5" to "Supplementary Figure S1" (Page 10, line 339); "Figure 6" to "Supplementary Figure S2" (Page 10, line 346); "Figure 7" to "Supplementary Figure S3" (Page 11, line 353); "Figure 8" to "Supplementary Figure S4" (Page 11, line 360).

Point 2: In fig. 3, I miss the data corresponding to the nucleic acids in different cells culture. These data are relevant for many biotechnological applications of microalgae

We agree with the reviewer’s comment that the data on nucleic acids in different cell cultures is relevant to many biotechnological applications of microalgae and will take this into account for our future research.
